# Barriers to Healthy Family Dinners and Preventing Child Obesity: Focus Group Discussions with Parents of 5-to-8-Year-Old Children

**DOI:** 10.3390/children10060952

**Published:** 2023-05-27

**Authors:** Blake L. Jones, Adam L. Orton, Spencer W. Tindall, Joshua T. Christensen, Osayamen Enosakhare, Keeley A. Russell, Anne-Marie Robins, Ana Larriviere-McCarl, Joseph Sandres, Braden Cox, Connor Thomas, Christina Reynolds

**Affiliations:** Department of Psychology, Brigham Young University, Provo, UT 84602, USA

**Keywords:** child obesity, family dinners, barriers, nutrition information, qualitative methodology, focus groups

## Abstract

Background: Although numerous physical and mental health benefits for children have been linked to family dinners, many families still do not have regular family meals together. This study sought to identify the barriers that keep families from having dinners together. Methods: We interviewed 42 parents of 5-to-8-year-old children in small focus groups to identify barriers and challenges that keep families from having healthy and consistent dinners together. Results: Parents reported the main barriers were time (e.g., time strain and overscheduling, mismatched schedules, long work hours, etc.), lack of meal planning or failure to follow plans, lack of skills (e.g., cooking skills or nutritional awareness), external factors (e.g., daycare, schools, or extended family, and competing with advertising), and food-related challenges (e.g., picky eating, food allergies). Parents also suggested potential solutions to overcome these barriers. Conclusions: Overall, parents had a desire to have family dinners with their children, but they felt that there are many barriers keeping them from establishing or maintaining consistent family mealtimes. Future research, as well as child obesity prevention and intervention efforts, should consider these barriers and suggested solutions in efforts to promote healthy and consistent family meals as a means of lowering the prevalence of childhood obesity.

## 1. Introduction

Child obesity is a nationwide epidemic. Recent estimates of obesity rates in the United States are approximately 20.3% for children 6 to 11 years old [1]. With obesity rates so high, early prevention and intervention efforts are crucial to prevent excess weight gain during childhood. Studies have shown that increased adiposity in young children is associated with increased metabolic risks (e.g., higher waist circumference, higher blood glucose levels, insulin resistance, etc.) in later childhood [2]. Those who suffer from childhood obesity also have an increased risk of later developing metabolic syndrome, cardiovascular disease, type 2 diabetes and its associated retinal and renal complications, nonalcoholic fatty liver disease, obstructive sleep apnea, polycystic ovarian syndrome, infertility, asthma, orthopedic complications, psychiatric disease, increased rates of cancer, and other health challenges [2,3,4]. Childhood obesity is a reliable predictor of adult obesity [5]. Therefore, the associations between obesity and negative health consequences emphasize the need for early prevention.

Some contributing factors to obesity include dietary decisions, lack of sleep, sedentary behaviors, and media use, among others [6,7,8,9,10,11]. As technology advances and sedentary lifestyles become more prevalent, changes need to be made in homes to help prevent parents and children from falling into these potentially high-risk lifestyle choices [12]. Familial routines seem to play a key role in both the problem of child obesity and its prevention. By addressing family routines, we may be able to assist families in fostering healthy daily habits [9,13]. In doing so, these routines could aid families in lowering the obesity rate for both children and adults. For example, in a longitudinal study on obesity interventions, the most successful long-term results came from parents teaching and practicing healthy eating behaviors with their children [14]. Epstein and colleagues found that more than 40% of children maintained their healthy habits for 10 years following the initial intervention. These findings suggest that when taught good eating behaviors from a young age, many children are able to retain those healthy eating habits over time. Thus, it would be beneficial to modify behaviors and teach healthy habits at an early age.

One of the key family routines that can influence a child’s weight and overall health is regularly gathering as a family for meals. Family meals have been associated with improved eating habits and food-related attitudes in children, greater consumption of fruits and vegetables, decreased risk of disordered eating, and improved psychosocial health, such as lower rates of depression and anxiety [15,16,17,18]. Family relationships, especially those between parents and children, also improve from eating together regularly. For example, children who had frequent family meals reported feeling more loved and connected to their parents than those who rarely or never ate with their families [19]. It is clear from these and many other findings that the benefits of eating as a family extend across many aspects of a child’s development.

The underlying causes of these benefits, however, are still debated in research. Some have pointed to the higher nutritional content of family meals compared to other meals as a possible explanation [15,16]. Others have suggested that the social dynamics and rituals of the family dinner environment, including conversation between parents and children and the involvement of children in food preparation, breed the greatest benefits to a child’s physical and emotional wellbeing [20,21]. Most likely, it is the multifaceted interaction of environmental factors, including positive experiences with family members, as well as the nutritional value of the food itself, that makes family dinners such formative experiences in a child’s life [18,22]. This may be especially true when evaluating links between family meals and obesity rates in children.

Although findings about the relationship between family meals and child obesity rates have not established a causal relationship [15,16], research continues to show a positive correlation between the frequency of family meals and lower obesity rates [15,18,22,23,24]. Because the rise in obesity prevalence has been associated with a reduction in the frequency of family meals in recent years [15,25,26], effective family meal practices have been suggested as a preventive factor against obesity [15,18,23,24,25,26,27]. Thus, family meals may contribute to the development of healthy eating habits within the family and may act as a powerful preventative factor in reducing child obesity rates. This makes sense from the perspective of social cognitive theory [28], which states that behavior is influenced by social relationships and modeling, such as the influence of a parent on a child. In the context of family meals, parents provide an important social referencing role and example for their children to learn healthy and appropriate mealtime behaviors that could help them lower their risk for obesity across their own development into adolescence and adulthood [29].

Furthermore, family mealtimes foster opportunities for positive interaction, bonding, and connection among family members, which decrease the risk for eating disorders [15,24,26,27]. Positive interactions during family mealtimes and eating with an adult family member can also reduce the likelihood of children being overweight or obese [15]. Planned, homemade family meals around the dinner table are often healthier than alternative options. In a systematic review of 54 studies and 11 review articles, researchers found a positive association between the frequency of family meals and intake of healthy foods (such as fruits, vegetables, dairy products, etc.) [15]. Likewise, family meal frequency and intake of unhealthy foods (such as fried food, sugary drinks, etc.) were inversely correlated [15]. A meta-analysis found that frequent family meals (three or more times per week) decreased the likelihood of childhood overweight (12%), unhealthy food intake (20%), and eating disorders (35%) in children [16]. Therefore, findings suggest that family meals support healthy eating as an effective and important obesity prevention strategy.

Despite the many benefits associated with family mealtimes, several barriers exist that make it difficult for families to have healthy and consistent meals. For example, time and schedule concerns have been identified as a barrier [30,31,32]. In addition to these challenges with time and schedules, other studies have reported barriers that include lack of meal planning [18,33], the cost of healthy food [31,34,35], lack of parental cooking ability and nutritional awareness [31,36,37,38,39], picky eating [40], food allergies [41,42,43], snacking [44,45], and some external factors such as advertisements and packaging [46,47,48,49].

Although these barriers make establishing and maintaining consistent and healthy family dinners a challenge for many parents, the links between family meals and the benefits, such as decreased child obesity risk, make the efforts worthwhile. Therefore, the goals of the current study were to identify the barriers that parents of school-aged children perceive would keep them from having family dinners together and to discover potential solutions to overcome these challenges. We hypothesized that parents would describe similar barriers as have been found in previous studies and that they would be able to work together in small focus groups to brainstorm potential ways to overcome these obstacles.

We chose this specific age of children because they are just old enough to start to help with planning and even preparing some parts of the meal, but they are young enough that they are not able to plan or prepare most dinners. This brings unique challenges into the family mealtime experience, and we wanted to focus on this particular age because they are old enough to participate in the mealtime experience, but the parents still have to carry the majority of the burden of planning and preparing meals and coordinating mealtime schedules. Furthermore, we chose to use a qualitative, focus group interview approach because it allowed us to ask follow-up questions and helped the parents to have rich group discussions with other parents who have children the same age as their child. This connection and discussion between the parents brought out more discussion of the potential barriers to family mealtimes and helped them to brainstorm together to identify potential solutions to overcome those barriers. These barriers and solutions were collected with the future design and implementation of a web-based tool in mind, with a focus on improving family meals as a child obesity prevention strategy.

## 2. Materials and Methods

### 2.1. Participants

The sample included 42 parents (mean age = 36.79 years; 36 mothers (86%) and 6 fathers) who had a child within the ages of 5 to 8 years old (mean age = 6.74 years; 26 girls (62%) and 16 boys). Participants were recruited from a mid-sized city in the Midwestern United States. Parents were primarily middle-class, with an average annual family income of $51,400 per year. The average education level was high overall, with 83% of the parents reporting that they were college graduates. The majority of parents identified as Caucasian (78.6%), with 9.5% being African American, 9.5% being Asian, and 2.4% being Hispanic. Most of the parents reported being currently married at the time of the study (76.2%), and they had an average of 2.7 children living in the home.

### 2.2. Measures and Procedure

This study was approved by the Institutional Review Board from the University of Illinois at Urbana-Champaign. Participants were recruited through various methods, including emails (sent to university listservs), flyers, word of mouth, and referrals from other participants. After hearing about the study, potential participants contacted the researchers via email or phone and completed a screening survey to ensure that they met the inclusion criteria for the study, which required the participants to have a child between the ages of 5 and 8 years old, to be able to speak and read in English, and to be able and willing to come to the university (blinded for review) lab for the study. If participants met the eligibility criteria, then researchers scheduled them to be in a small group that met on campus.

Data for this study were taken from a larger project. When participants arrived, they provided written informed consent. Then, they completed brief demographic surveys and participated in a nutrition website review (not included in the current study). After independently completing the demographics survey, participants reviewed four nutrition websites and completed a survey about how user-friendly and effective the websites were (these data were not included in the current study). Following the website reviews, the participants met in a small focus group with up to a total of six parents. The focus group questions were written as a semi-structured script and were given to a moderator, who led the focus group discussions. Questions included the following examples: please tell me about your usual family dinner routines; please describe the barriers and challenges that would keep families from being able to establish or maintain consistent family mealtimes; and please share some ideas for what might be possible solutions to overcoming these mealtime barriers (see Appendix A for list of questions that the moderator used). The moderator was then free to ask for clarifying information and ask follow-up questions as the groups discussed the topics of barriers to family meals and potential solutions. After the focus group discussions, which lasted approximately 30–60 min, participants were thanked for their participation and received $20 as compensation for their time.

### 2.3. Data Analysis

The researchers assessed the descriptive statistics to describe the participants’ characteristics and demographics. Next, all of the audio files of the focus groups were transcribed verbatim, and researchers coded the interviews to identify themes. Three trained coders independently went through each transcript to identify and label potential themes regarding family mealtime barriers, as well as solutions to overcome common mealtime barriers. Then, the coders met to reconcile any disagreements. After establishing reliability across the coders, the themes were divided into subthemes, and the count of parent comments was tallied manually for each subtheme for barriers and for solutions.

During data collection, the researchers were carefully monitoring the data to look for data saturation, which is one of the common ways to determine the final sample size in qualitative studies such as focus group studies [50,51]. Our initial plan was to collect data from about 50 parents. We were reaching saturation after collecting data from about 36 parents. We had already scheduled the remaining 6 families and wanted to ensure that we reached saturation, so we conducted the final focus group interviews to reach a total of 42 participating parents. After those final focus group interviews, we again checked and saw that we had reached data saturation.

## 3. Results

Using the quantitative surveys that parents completed prior to the focus group discussions, we assessed the demographic variables for the parents and their children (see Table 1 for demographics. In this table, we also reported on the child and parent weight status and BMI (body mass index), as well as the family mealtime variables from the survey.

Next, we assessed the qualitative data using the printed transcripts from the focus group interviews. Table 2 presents the themes, subthemes, and examples of parent quotations about each of those barriers to family mealtimes. This information came from the small focus group discussions that the 42 parents participated in for this study. Five main themes were identified as barriers to establishing and maintaining healthy family meals.

### 3.1. Time

The largest theme that parents reported was the issue of time. This included complications from parents’ work schedules getting in the way of family meals or parents working too much. Parents also mentioned being overscheduled so that the family experienced a time crunch. This led to having too many scheduled activities for the parents and children and not leaving enough time to sit down and eat together. Another concern with time had to do with meal preparation and convenience. Parents mentioned that when they are tired or want something easy that they have a hard time making good choices about family meals and nutrition. Finally, parents discussed the challenge with mismatched schedules. This was a problem for some families: not being too overscheduled, but just not having similar schedules and not being able to find times that line up to eat together.

### 3.2. Lack of Planning

This theme involved two parents not being able to create or maintain a plan for having family meals. The first subtheme related to parents not being sure how to plan family meals or be consistent in trying to plan. Parents talked about the stress of realizing that it was time for dinner and kids were hungry, but they did not have a plan for a meal or could not find ingredients in the moment and felt frustrated. They described the discomfort and panic of staring into the fridge or cupboards with no idea what to make for the meal, while the child is standing there wanting to eat right then and becoming upset. The second subtheme described situations where parents made a plan for the meal, but the meal fell apart due to various reasons such as last-minute conflicts or being tired or lazy.

### 3.3. Lack of Skills/Awareness

The third theme involved a lack of knowledge or skills to be able to create a healthy family meal. The main subtheme was a lack of nutritional knowledge. About half of the parents reported feeling that they had inadequate knowledge about basic nutrition because they had never been educated in this area. They talked about choosing unhealthy foods because they were not sure what the healthiest options were for their families. The other subtheme for this area was a lack of cooking skills or experience. Some parents stated that they were never taught how to cook healthy foods, or to cook anything, by their parents. This led to not having any confidence in meal planning or preparation, and the parents mentioned that they had to rely on fast foods, frozen foods, or other convenience foods that were not healthy.

### 3.4. External Factors

One of the themes that arose as a barrier to healthy meals was the influence of external factors. Parents discussed the frustration of trying to teach children about healthy foods when advertising and media tend to promote unhealthy food choices. They felt that this made it hard for them to convince their kids to want to eat healthier foods because they were influenced by the marketing of unhealthy foods more than healthy foods. The other external factors subtheme related to having others, including extended family and schools, push foods and eating habits that were not healthy. This made parents feel that others were compromising the efforts of the parents to help their kids want to eat healthy family meals.

### 3.5. Food-Related Challenges

The final theme was related to issues with the food itself. The first subtheme involved picky eating in the children, and parents felt that children being picky or fussy about various foods made it difficult to convince them to eat healthy foods in a family meal setting. This could be due to examples such as children refusing to eat what the rest of the family is eating and then making the parents feel as though they have to make multiple meal options to make everyone happy. The effect of this over time resulted in families who were not eating together and parents who were very frustrated with mealtimes. The other subtheme for food-related challenges involved children who have food allergies. This can also complicate mealtimes and lead to parents having to cook separate dishes for each family member, leading to another barrier that could decrease the frequency of family meals due to children’s health and safety or parents’ frustration at having to make multiple meal options.

### 3.6. Solutions to Overcome Mealtime Barriers

Finally, in Table 3 we present the solutions for overcoming the barriers that were offered by parents, along with examples of representative quotes from the parents about each type of solution. Some of the solutions included getting children more involved in mealtime planning and preparation. Parents described this solution as increasing the likelihood that children will be more invested in the meal. It also provides the additional benefit of teaching children about nutrition and cooking through firsthand experience. Parents also mentioned that user-friendly websites and internet resources could help parents to more effectively plan and prepare meals. Another suggested solution was to establish consistent parental control and rules to guide what types of foods should be eaten and included in family meals so that the children could focus on healthy foods. Additionally, parents mentioned the need to increase nutritional awareness and knowledge by using correct information to teach children about healthy foods and why they are important. Parents suggested that having more recipe ideas would be a helpful solution as well. They described the importance of making mealtimes a priority by putting this important routine into the family schedule to ensure it happens. This went along closely with the suggestion that creating better meal-planning habits would ensure success with family meals. Finally, they noted the importance of social support from technology and the community, among other solutions that were identified.

## 4. Discussion

This study focused on identifying some of the common barriers and challenges that parents of 5-to-8-year-old children face as they try to establish and maintain healthy family dinners. We then asked parents to help identify potential solutions for overcoming these barriers. Overall, parents reported many similar barriers to those cited in previous studies.

### 4.1. Common Barriers to Family Meals

The current study identified several barriers that prevent families from having healthy and consistent mealtime routines. These barriers supported previous literature that found that some of the big concerns are time and lack of planning, among others. Previous researchers also noted that time crunch, work schedules, overscheduling, and conflicting schedules can decrease the frequency of family meals. [30,32]. Time was a barrier that was mentioned several times throughout the interviewing process. Working parents often multitask before and during family mealtimes [30], and researchers found that 57% of parents purchased pre-packaged or processed meals due to time constraints [52]. Overscheduling was one of the most frequently described barriers related to time. According to national data in the United States, 57% of children in the United States are involved in one or more extracurricular activities [53]. The number of activities and events may cause parents to feel they lack the time and energy to cook or have healthy meals once they come home. Different schedules for each family member further complicate mealtime regularity. As one parent described it, the effect of overscheduling on healthy meal routines can lead to situations such as, “you come home, you’re tired and you don’t care. You just eat cereal.” Another commonly cited time barrier was parent work schedules, when one or both parents were scheduled to work during mealtimes. Parent work schedules, particularly among those parents whose jobs require shift work, placed greater strain on the regularity of family meals [54]. As work and extracurricular schedules become more demanding, the amount of time allocated for family meals within a household decreases. Another parent in the current study noted that parents struggle “in the very real world of situations that you deal with when parents potentially are in different places, and you have kids that have practices and schedules of their own”.

Meal planning and the cost of healthy food were also found to be challenges for parents. Planning was found to be a multifaceted barrier involving parents being unaware of how to plan meals and also not following through on their plans. Food planning may be an important barrier to consider, as researchers found that lack of planning was correlated with food insecurity within the home [33]. This is concerning because food insecurity has been associated with excessive weight gain, anemia, mental health risks, hospitalization, suicidal ideation, and many other health challenges [33,36,55,56]. Those that struggle with meal planning are less likely to have nutritious meals and might be more susceptible to certain health risks. The cost of healthy food was found to be a barrier to parents providing healthy food for their families, although parents were resourceful in this area [34]. Despite a caregiver’s ability to be innovative with food, cost continues to be an issue for many families, hindering families from providing balanced meals.

Parental cooking ability and nutritional awareness are two related barriers that may prevent families from having quality mealtimes. Parents who do not know how to cook will often purchase ready-made meals [39], leading to a decrease in their family’s consumption of fruits and vegetables [36,38]. Whether due to inability to cook or other reasons, parents are choosing to prepare more processed meals, resulting in the loss of fruits and vegetables in their children’s diets. Home cooking skills are on the decline, and ready-made meals have poor nutritional content when compared to their freshly made equivalents [36,37]. It seems that due to lack of cooking ability and efforts to save time, more parents are forgoing healthy, homemade meals for convenience foods.

Picky eating is a complex issue with various definitions in the literature, and it can be a serious barrier for some families. Researchers have noted that picky eating has been defined in various ways, such as not liking certain foods, limited intake, and resistance to the texture, appearance, or novelty of foods [40]. The prevalence of picky eating is high, with one longitudinal study finding that 39% of children were reported to have been picky eaters at some time within the ages of 2 to 11 [57]. Additionally, 14–50% of preschoolers were identified by parents as picky eaters [58]. This corresponds to the current study’s data, which indicate that several parents found that picky eating presented challenges for mealtimes. Some parents expressed that picky eating negatively impacted family mealtimes, with one parent describing gagging and screaming fits. Research has shown that picky eating can impact family meals [40] and can cause stress, distress, and conflict for families during mealtimes [59,60]. Researchers have also found that picky eating is a concern for parents. Researchers described parental concern as prevalent [57] and understood it to be a major concern [59]. Several parents communicated that they were concerned about picky eating, and their concern mainly revolved around their kids lacking fundamental nutrients in their diets and having nutritional deficiencies. Parental concerns about nutritional deficiencies were not unfounded, as picky eating has been linked to deficiencies in nutrition, as well as a poor-quality diet [61]. It is clear that picky eating is a common challenge for parents that impacts mealtimes, causes concern, and can present nutritional difficulties. Parents expressed that resources may be helpful in managing these challenges.

Food allergies can also be a barrier to family mealtimes, as they affect more than 6% of children [43]. The prevalence of food allergies in the pediatric population is also increasing [42], and allergic reactions can potentially be severe [41]. In the current study, parents stated that food allergies had caused their family to never eat out, or even to pull their child out of school because of food sensitivities and allergies. Food allergies have been found to impact family activities [43], as well as impair quality of life for families [35]. Food allergies can produce frequent mealtime concerns [42] and were discovered to increase meal preparation time [41] and increase overall economic costs for parents [35]. Because of the concern of parents regarding allergic reactions and the associated challenges and impacts of food allergies, more resources may be helpful in providing mealtime and related support for parents who have children with food allergies.

Parents communicated that certain external factors were a concern and created additional barriers to helping their kids eat healthy meals. The advertising of unhealthy foods targeted at children was one of those challenges. Additionally, worry about the quality of healthy food choices at school and not being able to control what their children ate while at school were also challenges. Parents in the current study expressed concern that their children were being bombarded by advertising for unhealthy foods. Studies have found that 72% of advertisements targeted at children were for low-nutrition foods [46], and 80.5% of all child-targeted advertisements the previous year were for foods within the lowest nutritional category [47]. These marketing techniques may be contributing to increased levels of childhood obesity [47,48]. One parent in the current study conveyed that they wanted to limit their child’s sugar intake. However, external factors may hinder this initiative and make that more difficult. For example, a study found that high-sugar cereals are the most-often-marketed product that kids see, and these advertisements contain unrealistic and contradictory information about healthy foods and behavior that can cause confusion for children [47]. Given these findings, it is clear that kids can be influenced by advertisers in choosing low-nutrition, calorie-dense foods. This is concerning because kids significantly influence the foods that are purchased at the store [62,63]. Approximately half of grocery shopping requests by children were for sweet items, and about half of those requests were fulfilled by parents, with branding and marketing accounting for about 28% of food choices [49]. Parents’ concern regarding child-targeted advertising as a challenge for healthy eating is widely supported by existing data.

Parents also reported concerns that their children may not be eating healthy foods and snacks at school, as well as concern regarding a lack of control over dietary choices while their children were at school. Data from 1977 to 2014 show that snacking for U.S. children has changed [44]. Consumption of salty snacks has doubled, and caloric intake from snacks increased 100% in that time [44]. As more calories come from snacks in a child’s diet, it is important that healthy snacks are available. In 2001, approximately 26% of elementary schools and 62% of middle schools had snacks available in vending machines [64]. Those numbers later decreased to 16% for elementary schools and 52% for middle schools in 2010 [65]. Despite this decrease, food from vending machines is available for many students, and availability increases as students age. Another study showed that schools were a powerful influence on student food choices, and that in grades six through eight, vending machine food options impacted the students’ diets positively or negatively depending on the selection of food in the machines [52]. The same study found that 83% of the food offered in vending machines was of poor nutritional value. Because of the prevalence of vending machines and the proportion of unhealthy foods among their offerings, parental concerns are legitimate. Through guidelines and various efforts, healthy snack availability seems to be improving, but efforts should be sustained, as deficiencies are still present [45]. Because children spend a significant amount of time at school, they are obtaining more calories from snacks. Schools should continue to examine healthier options and provide support for parents who are striving to give their kids a healthier diet.

### 4.2. Potential Solutions for Overcoming Barriers to Family Meals

One of the key contributions to this study was having parents brainstorm and identify potential solutions to overcome the barriers that they mentioned. Nine main categories or themes were identified from these parent suggestions. Each of these suggested solutions may show potential in overcoming the barriers to establishing and maintaining healthy and consistent family meals. For example, one of the suggestions was to get children more involved in the decision-making and preparation related to family meals. By the time children reach school age, they are still limited in some of their cooking and planning abilities. However, although it might not be safe or age-appropriate for a five-year-old child to cut and cook raw chicken, they can still help to measure out a cup of rice and a cup of water, or to wash broccoli or set the table. Parents in the current study noted that their 5-to-8-year-old children did not usually participate in mealtime planning or preparation and that it would likely slow down the parent during meal preparation. However, as they brainstormed, they noted that if they took the time to teach the children, the children would become more confident and competent in meal planning and preparation. Their idea was that over time the child would end up being a help to the parent during meal preparation instead of a hindrance. The other added benefit is that it would allow the parent to spend more time with the child and teach them directly, in addition to talking about their day at school and building their relationship. These ideas are supported by other researchers, who found that parents reported involving children in meal preparation could help increase family dinner frequency, particularly for dual-headed households [66]. Overall, this could help children feel more invested in the meal, as they are able to help plan and prepare the food and see where their food comes from.

Other solutions included focusing on better meal planning and establishing the family meal as a priority in the family routine and schedule. Because lack of meal planning can decrease both the frequency of family meals and the quality and health of the meals, it is important to encourage strategies to plan meals in advance. Parents in this study brainstormed ideas such as making a weekly meal calendar that the parent would plan on the weekend for the coming week. To combine this strategy with another solution, the parent would plan the week with the child present. This would allow the parent to strategize timing for meals that might fit better on days when cooking and preparation time were limited and would allow for the child to practice these important skills and be more invested in the meal schedule for the upcoming week. It would also allow parents to shop on the weekend to ensure that they have the ingredients they need for each meal that week. Parents even noted that they could include strategies such as cooking additional meat on Sunday that could be used in meals on other days in the week so that they could save time on those work nights. Previous researchers have suggested a similar intervention strategy of teaching parents to prepare quick and healthful meals that do not require a lot of time [67]. Making family meals a priority and taking time to plan in advance will likely go a long way in making sure that such a routine is followed.

Finally, parents suggested other strategies such as deciding on healthy rules for children’s foods and family meals, and using websites, social support, recipe ideas, and educational resources as means to increase their creativity and confidence. Parents mentioned that they sometimes lack knowledge about cooking and meal preparation and that by using these other resources they could increase their planning and cooking abilities, therefore making family meals more consistent, healthy, and enjoyable. It was clear in their interactions with the other parents in the focus groups that they wanted to connect to other parents who were facing the same mealtime challenges that they were. Some of the parents mentioned online forums, applications, and communities that they were part of that allowed them to discuss these barriers and find support from other parents who understood their situations. Perhaps one of the best solutions would be to increase the community support on applications and websites, as well as community-based connections, to help parents share ideas with others. This would give them a platform to find out what works for others and how to avoid common barriers that come up. Obtaining ideas from other parents may give them creative ways to try new meals or new strategies that could work for them as well. Additionally, it helps parents to know that they are not alone in their struggles to try to do everything well. They can relate to other busy parents who are trying to do their best to establish and maintain a healthy lifestyle for their families.

### 4.3. Limitations, Strengths, and Future Directions

The current study included several limitations. Although data were collected until we saw data saturation, the final sample size was still generally small and was fairly homogenous. This means that the data primarily described dual-headed households with parents who have college educations. People of every socioeconomic status, race, religion, and cultural background prepare meals for themselves and for their children; however, the study included a limited scope of diverse persons, mainly in that it included those who had the time and resources necessary to participate in the study. Also, the study only included parents living in the Midwestern United States, so there are likely many different perspectives in other areas of the United States and in other countries around the world. Future researchers should continue to examine these issues in larger and more diverse samples to help increase the external validity and applicability of the results. They should also consider comparing differences in family mealtime barriers across countries and cultures to look for common experiences and diversity, as this will provide a richer discussion of how family mealtimes matter throughout the world. We can also learn more from the successful strategies of parents around the world.

Additionally, having more participants and studies in other areas would likely further validate these findings. The method used to divide participants into groups may have also influenced the type and amount of information received in response to questions, as the number of participants placed in each group did decrease as the study progressed. The early focus groups were larger, containing five to six participants, whereas the last groups were smaller. This could have prevented vocal participants from voicing opinions in the first groups, or possibly not provided enough priming to allow participants to share all their opinions in the latter groups. We also used two different facilitators during the course of this study, which could have changed the information shared during the interviews if some topics were framed differently by the two different facilitators who used a semi-structured interview process.

While there were multiple areas for improvement, the strengths of this research include the data collection procedure and the standardization of the questions asked to each focus group, as well as the data collection procedure during the group interviews. Information was gathered directly from parents, who met in small groups where they were encouraged to talk and share their experiences without being rushed. This led to extensive, rich, qualitative, and useful data. Children mature and change rapidly, and the narrow range of ages included in this research led to particular insight into a specific group of children that are at a critical age for obesity prevention and are just old enough to start helping with meal planning and meal preparation.

Future studies and intervention efforts regarding barriers to helping children and parents establish and maintain healthy mealtime habits should consider several areas of focus. First, it is important to understand the common barriers that families face in the home feeding environment. Sometimes researchers devise great intervention strategies in theory, but they do not succeed in practice because parents are dealing with real challenges that keep them from implementing those interventions effectively. Second, it is important to consider how barriers may differ across different populations. Other populations should also be studied so that barriers specific to cultures, age groups, and economic statuses can be identified, thus resulting in more specific and effective intervention methods. Further research should be focused on detailing these barriers with more specificity across different child ages and for different ethnic and cultural groups in order to more fully understand them. This focus will allow the interventions to be more tailored to the specific population of interest so that they can be more effective. Third, future research should also aim to create quantitative and objective measures to assess these barriers, allowing medical professionals and clinicians the ability to assign their patients to proper, evidence-based interventions that are specific to their clients. Finally, future researchers should examine and test the effectiveness of the potential solutions that the parents identified in this study to assess their effectiveness in increasing consistent and healthy family dinners. By identifying which solution ideas are effective, researchers will be able to promote those solutions to parents and include them in child obesity prevention and intervention efforts that target family mealtimes.

## 5. Conclusions

As parents and intervention researchers are able to identify and address the barriers that hinder family meals and keep them from having family dinners together, they will be able to increase the likelihood that families will establish and maintain regular, healthy family mealtimes. This will also allow children to become more involved in the meal planning and meal preparation in their families, which will teach them these important life skills that will allow them to prioritize and maintain their own healthy mealtime routines on their own and with their future families. Children will learn to recognize that family mealtimes are important and will have the knowledge and resources to make family mealtimes a priority and apply these skills in their own daily routines. By having children become more involved in meal planning and meal preparation, they will learn where their food comes from. Additionally, as they gain nutritional knowledge and awareness, they will also have the opportunity to practice their own meal planning and cooking skills. Among the ultimate goals of increasing family mealtimes is that children will obtain the added benefits that are associated with family mealtimes, including a decreased risk for child obesity and other health problems associated with child obesity.

## Figures and Tables

**Table 1 children-10-00952-t001:** Demographic Information and Family Mealtime Climate Variables (*n* = 42).

Individual Demographic Variables	Child Mean *(sd)*	Parent Mean *(sd)*
**Age** (in years)	6.72 (1.07)	36.79 (6.34)
Sex	Girls = 26 (61.9%) Boys = 16 (38.1%)	Mothers = 36 (85.7%) Fathers = 6 (14.3%)
Child BMI Percentile ^a^	60.39 (32.44)	--
Parent BMI	--	24.92 (5.69)
Healthy Weight Status %	64.9%	61.9%
Overweight or Obese %	35.1%	38.1%
Race − Non-Hispanic White − Non-Hispanic Black − Asian/Pacific Islander − Hispanic	73.8% 9.5% 14.3% 2.4%	78.6% 9.5% 9.5% 2.4%
**Family Demographic Variables and Mealtime Climate Questions**		**Mean (*sd*) Or %**
Parent Education Level − High School Diploma/GED − Some College − College Degree − Graduate or Professional Degree		4.8% 11.9% 42.9% 40.5%
Family Income (mean)		$51,400 ($30,454)
Marital Status − Married − Single − Divorced − Widowed		78.6% 11.9% 7.1% 2.4%
Total Number of People in the Home		4.70 (1.51)
Family Mealtime Climate Questions “Our family regularly eats the main meal together” − Never/Seldom − Sometimes − Often − Always		4.8% 14.3% 38.1% 42.9%
“In our family, mealtime is planned in advance” − Never/Seldom − Sometimes − Often − Always		11.9% 14.3% 45.2% 28.6%
“In our family, we feel it is important that we eat together.” − Never/Seldom − Sometimes − Often − Always		2.4% 11.9% 21.4% 64.3%
“In our family, everyone is expected to be home for the main meal.” − Never/Seldom − Sometimes − Often − Always		14.3% 9.5% 38.1% 38.1%
“In our family, everyone has a specific role or job to do (during the meal).” − Never/Seldom − Sometimes − Often − Always		30.9% 35.7% 28.6% 4.8%
“In our family, mealtime is flexible; people eat whenever they want.” (This item is reverse coded when calculating the mealtime climate scale.) − Never/Seldom − Sometimes − Often − Always		73.8% 19.0% 2.4% 4.8%

Note: ^a^—Child BMI is more appropriately presented as BMI-Percentile because this variable factors in their age and sex. Adult BMI score is used for the parents because after age 19 the BMI score itself is used to calculate weight status (Child = above 85th percentile is considered overweight, above 95th percentile is considered obese. Adult = BMI score over 25.0 is considered overweight, and over 30.0 is considered obese.).

**Table 2 children-10-00952-t002:** Themes and Subthemes of Parent-Reported Barriers to Establishing and Maintaining Healthy Family Meals (*n* = 42).

Barrier Theme: Subtheme	Description (# of Comments)	Representative Quotations
Time: Parental Work Schedules	Parents’ work schedules conflict with mealtimes/healthy mealtime habits. (12)	Something that my husband and I have struggled with, because we both work full time, is ourselves maintaining healthy routines… whether that’s eating at regular times because a lot of times I’m making food for them, and then I don’t want to eat what they’re eating so then I’m making something else for me, so then I’m not sitting down with them. (non-Hispanic white mother of boy) I think time is the biggest challenge because if I get off work... at 4:30 p.m., pick up my kids at by 5:00 p.m., 5:15 p.m. get them supper. So we’re done by 6:00 p.m. but that’s still only an hour and a half before we start our bedtime routine…[and] my husband doesn’t get home until 6:30 p.m. So from 6:30–7:30 p.m. is the only hour he has with them. (non-Hispanic white mother of boy)
Time: Overscheduling/ Time Crunch	Overscheduling of children/parents creates a barrier. (22)	We’d like to be more active, but with each of us at a different age and interest level, I don’t want us all doing something a different night of the week (non-Hispanic white mother of girl) But this is what gives rights to the over scheduling because... we need to get them in soccer, and we need to get them in gymnastics (non-Hispanic white mother of boy) My child, I feel like has a longer work day then I do…I drop her off at school at the before program and then she’s still at school for the after program. So she’s there from like 7:45 a.m. to sometimes 6:00 p.m. (non-Hispanic white mother of boy)
Time: Convenience	It is more convenient time-wise to not teach healthy habits. (13)	We made bad choices in how to make life easier and some of those choices that we made…have actually been detrimental to our health in the end. (non-Hispanic white mother of girl) And you come home, you’re tired and you don’t care. You just eat cereal. (non-Hispanic white mother of girl)
Time: Mis-Matched Schedules	One parent works an unusual shift, and thus work schedules are mismatched. (10)	It’s tricky…time. Lack of time with two parents working. (non-Hispanic white mother of boy) We have a breakfast meal time routine, we do have that. I try to keep it because it’s the main meal time… but for our family it is because my husband often works in the evenings and so it’s just me and the girls. (non-Hispanic white mother of boy) In the very real world of situations that you deal with when parents potentially are in different places, and you have kids that have practices and schedules of their own, and how you pull it all together and you make that work in a way that supports healthy growth and development. (non-Hispanic white mother of girl)
Lack of Planning: How to Plan?	Parents are unsure how to plan. (12)	We don’t ever meal plan...I just buy a variety of things that might be helpful throughout the week to me. (non-Hispanic white mother of boy) If I could find a really good meal planning for a week… that would be awesome because sometimes they’re too much work. (non-Hispanic white mother of girl)
Lack of Planning: Failed Plans	Parents make plans, but they fail to complete them. (4)	It’s like, Okay, I’m tired. My plan was to make this dinner but now I got off work late and it’s 6:30, so let’s have a canned bowl of ravioli, or, Let’s have a bowl of cereal or something fast. (non-Hispanic white mother of girl)
Skills/Awareness: Cooking Skills	Parents lack cooking skills. (5)	Well, now we’ve got a whole group of people because of… decisions… who have lost the skills of how to even prepare simple things and so those skills on how to prepare raw food versus how just to make a boxed meal we’ve lost the skills. (non-Hispanic white mother of girl) I’d say our biggest challenge is cooking with…as much whole foods as possible, staying away from processed foods...Using that in moderation, not relying on the quick foods to cook with. (non-Hispanic white mother of girl) My daughter, who’s five…enjoys helping out in the kitchen, but sometimes it’s more challenging to find age-appropriate things for her to do besides setting the table. (single, non-Hispanic white mother of girl)
Skills/Awareness: Nutritional Factors	Parents lack nutritional awareness about foods. (21)	I believe a barrier is how the parents are raised...I know...entire kingdoms of food they [parents] won’t even touch, and therefore that’s gonna get imprinted upon your kids...So I think parental experience is a large barrier. (non-Hispanic white father of boy) Child obesity rates have tripled in the last decade and adult obesity rates are continuing to sky-rocket. We have a problem of people not making healthy food choices. (non-Hispanic white father of girl)
External Factors: Advertising	Advertisements create difficulty in teaching healthy eating habits. (10)	[I think the biggest challenge is] limiting sugar intake… because, I feel like…my kids, [say] “May I have a treat?” every time we turn around whether it’s at the babysitter’s or whatever event we go to. Everybody is marketing them with a treat, and it’s really hard to be on the other end. (non-Hispanic white mother of boy) I think the biggest barrier is that we’re told from the media what’s healthy and so we believe it because it’s on the TV or it’s on the Internet and that’s not what’s good for us. (non-Hispanic white mother of girl) All of the cereal boxes in the cereal aisle have characters and bright colors [which are] visually appealing to a child. They want the pretty red colorful box so, I kinda feel like there’s some good old marketing tactics there that the healthy stuff could work on. (non-Hispanic white father of girl)
External Factors: School/ Daycare/Extended Family	Outside sources create difficulties when teaching healthy eating habits. (8)	I’m not happy about that [what the school feeds my child], I’m not sure I feel like I have control to make improvements, but I think Pop Tarts really don’t suffice for a good breakfast. It’s like a candy bar for breakfast…I drop my child off early and so they have breakfast at school, and so I don’t know if he’s eaten [healthily] or not. (non-Hispanic white mother of boy) Her older brother also can be somewhat difficult in terms of his attitudes and perceptions about nutrition and food. She definitely picks [it up] so she’s watching him and listening to him and if he’s making faces and saying “yuck, I don’t like this”, then she’s likely to pick up on that and kind of mimic him. (single, non-Hispanic white mother of girl)
Food Related Challenges: Picky Eating	Parents find difficulty in teaching healthy eating habits to picky eaters. (13)	I’m just worrying about health and that she’s getting the right nutrients to grow…The doctor even was worried about her gaining weight this year, so we always worry about finding the things that she can eat that will give her everything that she needs. Hispanic mother of girl) My younger child is just all about…chicken McNuggets…wants the sweets…the ice cream… So then we have this clash for what I’m putting on the table is being rejected. (widowed non-Hispanic white mother of girl) … [My child] tends to be a little bit more of a picky eater and fussy. (single, non-Hispanic white mother of girl)
Food Related Challenges: Food Allergies	Child has food allergies that limit meal options. (7)	Both of my kids have food allergies…it’s a different situation, but they’re involved in the cooking and all because of that…their food allergies are severe enough that we pulled them out of school and we are homeschooling them. (non-Hispanic white mother of boy) [My child] had a dairy thing and a peanut thing but I don’t think it was a true allergy, or it was just more like a reactive type thing. (single, non-Hispanic white mother of girl)

**Table 3 children-10-00952-t003:** Parent Suggestions for Solutions to Healthier and More Consistent Family Mealtimes (*n* = 42).

Parent Suggestion Theme	Description/ (# of Comments)	Representative Quotations
Child involvement/child decision	Children should have more choice and selection regarding foods and should be more involved in both meal planning and meal preparation. (34)	We still have meals planned out a week in advance. The kids play a role in that. Some things are simple for the kids, like…chopping vegetables, or…I make pizza once a week, and the kids help to roll out the dough and to put their stuff on there. So it’s empowering them to understand that food can be something that they play an active role and…they can really decide. (non-Hispanic white father of girl) Because when they do it themselves, they are all about it. They are in it. They are invested in it. They want to eat it. They want to try it much more so than when I just put it in front of them. (widowed non-Hispanic white mother of girl) The one I always let them control is whether or not they’re full... she’s eaten…and [will] say, “I’m full, I’m done.” I say fine, I’m not going to make you eat more. Because… I’ll eat regardless whether I’m actually hungry…. Why am I eating? Actually I’m full, I don’t need to eat.” And getting them to understand that because, kids are natural at that, they don’t have to be taught that, they start out that way. (non-Hispanic white mother of girl)
Website suggestions	Nutrition and health websites should contain user-friendly features that make meal planning, nutrition education, and meal preparation easy for parents and children. (40)	I know all of my friends would use it because when we’re trying to figure out what are we gonna eat this week, how are we gonna feed the family, working moms and non-working moms, it would be great if there was a website to help us build up something healthy that suits the whole family…. healthy meal planning ideas. (non-Hispanic white mother of girl) An ingredients search tool where you can put in the ingredients that you have and it will come back with some recipes that incorporate those ingredients you might already have on hand… And keeping it geared towards recipes that might be done in less than 30 min, things that can be prepared quickly that the kids might be able to get involved with at some level. (single, non-Hispanic white mother of girl)
Parental control/rules	Parents should guide children to make healthy choices and have rules in place about what the children can and should eat. (10)	Usually my kids choose like hot dogs and pizza and macaroni and cheese and things like that. And every once in a while I’ll give in, but I’ll try to make those types of meals healthier…. I try to make my own pizzas, and that way I’m choosing what’s going in the dough, and I’m choosing how much salt is in the sauce and things like that. (non-Hispanic white mother of boy) I think children should expect to not own the decision to what they’re eating…. there’s a balance there, but I just think that in our society in general… parents are not willing to step up and just say, ‘I’m the parent, and I need to decide what’s best for you,’ and I think that definitely gets into food choices. (non-Hispanic white father of boy)
Education/ external sources	Meals and resources should provide educational opportunities for children to learn more about healthy food choices and habits. (33)	I rely on government websites for nutrition information, for the latest clinical research that’s going on in allergy treatment. I trust the government, and I rely on those websites a lot. I like people in the forums, but they are not as reliable as these websites which provide hardcore evidence based on research that’s going on. (non-Hispanic white mother of boy) I like to teach them about the good that is in their food and what it does for them…. I try to tell about each of the components, like, ‘You like oranges? There’s vitamin C in that, and it does all this for you...’ So, connecting the dots for kids helps a lot. (single, non-Hispanic white mother of 6-year-old girl) So again, it’s validating these external sources other than what they are just getting in the home… They are getting to that age where external sources and peer sources are becoming more important, and that information seems to resonate more with them. (single, non-Hispanic white mother of 7-year-old girl)
Recipe ideas	Ideas for meals and foods that are healthy, fast to prepare, balanced, cost-effective, and/or convenient to make. (15)	I think coming up with some alternatives to these snacks because, I think all children are somewhat programmed to just want sweets, and trying to have alternative snacks that they could use to supplement for that kind of thing, is something that I would like to incorporate more. (non-Hispanic white father of boy) I want…meal ideas. I liked that a lot on the websites we saw. There were meal ideas. There were healthy, creative… different recipes… I like ideas like that. (non-Hispanic white mother of boy)
Routine/ schedule	The importance of making mealtimes an established and consistent routine for the family (13)	I realized that, for us, routine is everything. You get off the routine; my youngest will eat all day long. He is a grazer and he just, he will eat, so I have to, it’s not ten o’clock. You cannot have your snack yet and if you’re still hungry, then you should have eaten your breakfast. So we’re very strict about when they can eat, not necessarily what they can eat, but when. (Asian mother of girl) We make dinner time a routine. We feel that’s a really important time in our house to have our family together at the dinner table at that time. We always eat around the same time. I will plan the meals for the whole entire week and… we [keep] up that routine. (non-Hispanic white mother of boy)
Plan/ prepare	Planning meals out in advance so that they are healthy and having healthy snack options ready for hungry kids (17)	We try to prepare, or I try so hard because I’m bad at this if I don’t, but we try to prepare snacks ahead of time so that if people are hungry it’s carrots and it’s not something that would not be so healthy in that moment. (Asian mother of girl) I’m not really good at meal planning, but I’m wondering if it would help my kids to decide. We have a list of meals that are their favorites, and usually on Sundays we’ll plan the week for at least what they’re cooking... if there was like a website where they could do...meal planning for the week, and then they do a printable shopping list. (African American mother of boy) Every weekend we sit down and we make the week’s meal plans that we can make the grocery list based on that and we factor our leftovers and if we have to go out one night. (non-Hispanic white mother of girl)
Social support	Using social support and technology to help with healthy meal ideas for families (4)	Take advantage of things that are already out there…What comes to mind is My Fitness Pal, or Map My Run…there’s a lot of social networking involved where when you enter a food it gets put into the database. So everyone is contributing. (Asian father of boy) I’m heavily involved in...local Yahoo! groups that are for parents, families... And you can go on there and discuss any topics you want, about parenting, about your kids… And you meet people, you go to play groups, play dates, go to the parks.... then you have those people who could watch your kids when you need them to... And then also nationally on part of other forums where I get on, I say, ‘I’m having this problem, what do you guys do?’ And I trust those people because they’re like-minded people, I’ve gotten to know them over years. (non-Hispanic white mother of Asian girl)
Other	Any other ideas that would be helpful for increasing the healthy and consistency of child and family meals (14)	Our family diet is basically the “Rainbow diet;” the more colors in the foods, the better. We kind of avoid the brown, the white, processed. And I think the kids respond to that... (non-Hispanic white father of girl) I saw on one of the websites there was a link to like books, kids’ books about the adventures of some broccoli or something. Those look totally fun... my kids love books and they really connect with stories and that makes things exciting for them. So if we end up at the dinner table and we can make up a story about our broccoli… they are more likely to eat it. (non-Hispanic white mother of girl) I once visited [my child’s] school during lunchtime, and they just don’t have enough time to eat…. it’s a very short lunch period, and if you’re talking and all that you’re gonna waste your time. So at school I would love to see… a longer lunch period, so allow kids more time to eat and…digest. (non-Hispanic white mother of girl)

## Data Availability

This was a small grant-funded study, and the data have not been made publicly available. If you are interested in obtaining or accessing any of the data, please contact the authors.

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
