# Peer review of "Barriers to Healthy Family Dinners and Preventing Child Obesity: Focus Group Discussions with Parents of 5-to-8-Year-Old Children"

_children, 2023, doi:10.3390/children10060952_

Round 1

Reviewer 1 Report

This report provides a summary of the difficulties in finding time to eat meals with family members.

This report is unacceptable as a thesis because it is the result of a roundtable discussion rather than a research direction.

It is also regionally specific and may not be a globally common and desirable form of mealtime.

It is necessary to specify in which region the importance of family meals is emphasized.

Methods

Line 112- The number of participants needs to be stated in the results section.

Line 123- "IRB" can be spelled out.

Line 138- All questions need to be included in the appendices (consider including them in the appendices).

Results

Please provide a table showing the background of the participants (including age of children, family structure, lifestyle, health literacy, educational history, etc.)

It is necessary to scientifically examine which components are barriers to eating together as a whole family, e.g., by text mining.

In particular, it is essential to identify the contribution of each factor (especially those that are major obstacles).

It is necessary to gather families who actually eat meals with their families and those who do not, and compare all factors.

For the subjects of this study, the current eating situation (do they eat with their families) and the situation of their children (obese or not) would be necessary information.

Author Response

Thank you for your feedback and the opportunity to revise our manuscript. We really appreciate your comments, concerns and feedback and feel that we have addressed the concerns as much as possible. We believe that these changes have increased the clarity and quality of the paper and we appreciate your help in this process.

Reviewer #1 (our responses are in bold)

This report provides a summary of the difficulties in finding time to eat meals with family members.

            Response: Thank you for your time and helpful suggestions.

This report is unacceptable as a thesis because it is the result of a roundtable discussion rather than a research direction.

            Response: This study was designed to be an exploratory study using a mixed methods approach, with primarily qualitative data from the focus group interviews and some quantitative survey data. Focus group studies like this are commonly used when we are trying to identify themes and responses from a group of interest. In this case, we wanted to learn more about the specific barriers that are faced by parents of children who are 5-to-8-years-old because these are kids that are just getting old enough to start helping in mealtime planning and preparation, but are too young to cook, plan, or prepare most dinners on their own. We wanted to talk to these parents in focus groups because it is an effective methodology in getting them to open up about real-life concerns and challenges with other parents who might be experiencing similar things, and is very helpful at allowing them to brainstorm ideas that are often helped by the comments of other parents who are experiencing similar challenges with kids being the same age. By using these qualitative focus group interviews, we were able to have parents expand on their comments and our moderator could ask follow-up questions to identify more rich details about the family mealtime experience, barriers, and possible solutions to overcome those barriers. To address these issues, we also added some notes about why we chose this population and methodology to the end of the introduction (just before the method section).

It is also regionally specific and may not be a globally common and desirable form of mealtime.

            Response: Thank you for this comment. This data was collected in the Midwestern United States, and as such this presents a limitation as it only represents a homogenous group in one area of one country. We have now added this limitation to the discussion in the top of the limitations section.

It is necessary to specify in which region the importance of family meals is emphasized.

            Response: This is a helpful comment. This study only recruited participants from the Midwestern U.S. in the state of Illinois. It was done in person and conducted locally where the researchers resided at the time of the study. Because of those factors, the study did not ask about parental perceptions of parents outside of that area, but instead focused on the perspectives of only the parents that participated. This limits the scope of the study to region that it took place in. However, after analyzing the data and themes, we noticed that many of the barriers were similar to previous studies about mealtime barriers that were conducted in other states and in other countries. This is a good reason to continue to promote this work worldwide, so that we can compare family mealtime experiences and barriers across cultures and countries. We have added more comments about the need for this work to be conducted worldwide (in the limitations section). Thank you again for this important suggestion.

 Methods

Line 112- The number of participants needs to be stated in the results section.

            Response: Thank you. We have added a line in the results section again reiterating that there were 42 total participants. It also notes the number of parents in the method section (in the usual section of the participants), in all of the tables, and in the abstract. In line 112, it had already stated that there were 42 parents. We left this in the paper.

Line 123- "IRB" can be spelled out.

            Response: Thank you for this suggestion. We changed it to say “Institutional Review Board from the University of Illinois at Urbana-Champaign.”

Line 138- All questions need to be included in the appendices (consider including them in the appendices).

            Response: Thank you. We added a note about seeing the appendix for the full list of questions that the focus group moderator asked and we added the questions to the appendix.

Results

Please provide a table showing the background of the participants (including age of children, family structure, lifestyle, health literacy, educational history, etc.)

            Response: This is an excellent suggestion! Thank you. We added several of the demographic variables as well as more information about the children and families to the new Table 1. This was a very helpful addition to the paper and will hopefully help the reader to understand this population better. Thank you again!

It is necessary to scientifically examine which components are barriers to eating together as a whole family, e.g., by text mining. In particular, it is essential to identify the contribution of each factor (especially those that are major obstacles).

            Response: This is essentially what we did in this manuscript. We had multiple trained researchers go through the printed (transcribed) transcripts of the focus group interviews and code for themes in the text and then do sum score counting to see which of the comments/themes were the most influential barriers. In Table 1, we present the sum scores for which themes were mentioned. In each of the sections, along with the representative quotes from the parents, we give the sum score for number of comments in parentheses (and it is bolded). We have seen similar tables in other publications and tried to follow this method for giving sum scores with the qualitative data. If there is a better way to represent these sum scores, we are happy to consider other suggestions.

It is necessary to gather families who actually eat meals with their families and those who do not, and compare all factors.

Response: Thank you for this suggestion. It pointed out that we had not included the frequency of family dinners for the families that completed the interviews. We have now added this information to Table 1. It shows that the majority of families said that they always (n = 18) or often (n = 16) had family dinners together, but that there were some families that only had dinners sometimes or very rarely (n = 8). This gave us a balance of families with different experiences and perspectives on what barriers make it difficult to regularly eat dinner together. Some families bring the perspective that they have found at least some practical and successful ways of overcoming barriers to eat as a family because they always find a way to make this routine happen. Other families try to have family dinners together as often as possible, but they run into barriers that keep them from their goal of eating together each night. And finally, some families may face extra or more challenging barriers that prevent them eating together. We feel that with these varying perspectives, it gave us a more balanced understanding of how family mealtime barriers are experienced by different families. In Table 1, we also note the different mealtime priority questions such as whether the family feels that it is important to eat together, if everyone has a role in family meals, and if everyone is expected to be home together for family meals.

Again, we are very thankful that the reviewer brought up this point so we could add this data to Table 1 and to the results section as it offers a meaningful way to highlight that varying perspectives of parents help this discussion of mealtime barriers.

For the subjects of this study, the current eating situation (do they eat with their families) and the situation of their children (obese or not) would be necessary information.

            Response: Thank you for these comments as well. We have added the information about family dinner frequency and obesity of the children and parents to Table 1 and the results section. We agree that these are helpful points of information for this study.

Reviewer 2 Report

            Thank you very much for allowing me to review the article titled "Barriers to healthy family dinners and preventing child obesity: Focus group discussions with parents of 5-to-8-year-old children" (children-2424337).The hypothesis is that parents would describe similar barriers to those found in previous studies, and that they would be able to collaborate in small focus groups to generate potential strategies to overcome these obstacles. The objective is to identify the barriers perceived by parents of school-aged children that hinder their ability to have family dinners together and to explore possible solutions to address these challenges.Comments:This is a qualitative study that provides highly interesting information.Has a sample size calculation been conducted prior to the study? How many parents have accepted or declined to participate in the study? What are the reasons for declining participation? How was the recruitment process conducted, and what type of sampling was employed?

Author Response

Reviewer #2 (our responses are in bold)

Thank you very much for allowing me to review the article titled "Barriers to healthy family dinners and preventing child obesity: Focus group discussions with parents of 5-to-8-year-old children" (children-2424337). The hypothesis is that parents would describe similar barriers to those found in previous studies, and that they would be able to collaborate in small focus groups to generate potential strategies to overcome these obstacles. The objective is to identify the barriers perceived by parents of school-aged children that hinder their ability to have family dinners together and to explore possible solutions to address these challenges.

Comments: This is a qualitative study that provides highly interesting information.

            Response: Thank you for your time and your careful review and positive feedback about the study in general.

Has a sample size calculation been conducted prior to the study?

            Response: In other studies, we generally calculate the needed sample size based on the expected effect size and number of variables, etc. However, because this was an exploratory study that used qualitative methodology, the sample size was expected to be about 50 parents coming in to participate in the focus groups. During the later stages of our data collection, we were carefully monitoring the interview transcripts to look for data saturation. With qualitative data and focus group data, data saturation often guides the researchers to know when to stop data collection because this shows that new ideas are no longer coming up in the parents’ comments. We started to see data saturation after about 36 participants, but because the remaining 6 parents were already scheduled, we decided to follow through and collect that data as well and watch carefully for any new comments/themes. Again, we did not see anything new in the final 6 participants so we made the decision to stop data collection with a sample size of 42 parents in this case. We realized, because of your comment here, that we had not included this important information in the paper. Thank you for catching that omission! We have now added those details to the data analysis part of the Method section. For more information about data saturation, articles such as Morse, 2015; or Saunders et al., 2017 are particularly helpful.

How many parents have accepted or declined to participate in the study?

            Response: We had a total of 42 participants. Only one other parent contacted us and went through the screening interview and scheduled to participate in a focus group. But the parent did not show up and declined to reschedule because they were too busy (this is mentioned and given more detail in the comments below to address the next questions from the reviewer).

What are the reasons for declining participation?

            Response: Once families contacted us to ask about participation, all of those families met the screening criteria for inclusion and participated except for 1 family that did not show up to their scheduled focus group. They decided that they were too busy and when we tried to reschedule with them, they told us that it was just a very busy time for them so they opted to not participate. But of those that started the study, we did not have anyone decline participation or drop out of the study before they completed each part.

How was the recruitment process conducted, and what type of sampling was employed?

            Response: This study used a mix of having a convenience sample with a snowball (referral) sampling approach. We initially reached out to local parents in the community through emails to university employees, flyers posted in grocery stores and other public locations (e.g., libraries, churches, family and community centers), and word of mouth. We then asked participants to share the information about the study with any friends, neighbors, or relatives that might have a child between the ages of 5 and 8 who might be interested in participating. We share some of this information in the Measures and Procedure section on page 4 at the top. Ultimately, it was a convenience sample and the participants reached out to us to show interest, and then we did a screening interview over the phone to make sure that they had a child between 5-to-8-years-old and that they were able to speak and read English and were willing to come to University of Illinois campus. If they met those criteria, then they were able to participate and we scheduled them into a focus group with other parents.

Because it was a convenience sample and we used a variety of methods to post and inform potential participants about the study, we have no way of knowing how many people decided not to contact us. Of those that contacted us, all of them (except 1) participated and completed the study. There was one family that signed up for a focus group and then never showed up. We called them back and they said that after thinking about it that they were too busy at the time so they opted not to participate at all. We felt that this high number of people who actually completed the study was likely due to having very clear information about the study and inclusion criteria on the flyers. Also, given the topic, a lot of parents told us that they were very interested to talk to other parents about family mealtime barriers and solutions.